# Acceptance and Commitment Therapy for Muscle Disease (ACTMus): protocol for a two-arm randomised controlled trial of a brief guided self-help ACT programme for improving quality of life in people with muscle diseases

Michael R Rose,[1] Sam Norton,[2] Chiara Vari,[3] Victoria Edwards,[3] Lance McCracken,[2] Christopher D Graham,[4] Aleksandar Radunovic,[5] Trudie Chalder[3]

For numbered affiliations see end of article.

**Correspondence to**
Dr Michael R Rose;
m.r.rose@kcl.ac.uk

## ABSTRACT

**Introduction** In adults, muscle disease (MD) is often a chronic long-term condition with no definitive cure. It causes wasting and weakness of the muscles resulting in a progressive decline in mobility, alongside other symptoms, and is typically associated with reduced quality of life (QoL). Previous research suggests that a psychological intervention, and in particular Acceptance and Commitment Therapy (ACT), may help improve QoL in MD. ACT is a newer type of cognitive behavioural treatment that aims to improve QoL by virtue of improvement in a process called psychological flexibility. The primary aim of this randomised controlled trial (RCT) is to evaluate the efficacy of a guided self-help ACT programme for improving QoL in people with MD. Main secondary outcomes are mood, symptom impact, work and social adjustment and function at 9-week follow-up.

**Methods and analysis** Acceptance and Commitment Therapy for Muscle Disease is an assessor-blind, multicentre, two-armed, parallel-group RCT to assess the efficacy of ACT plus standard medical care (SMC) compared with SMC alone. Individuals with a diagnosis of one of four specific MDs, with a duration of at least 6 months and with mild to moderate anxiety or depression (Hospital Anxiety and Depression Scale score ≥8) will be recruited from UK-based MD clinics and MD patient support organisations. Participants will be randomised to either ACT plus SMC or SMC alone by an independent randomisation service. Participants will be followed up at 3, 6 and 9 weeks. Analysis will be intention to treat, conducted by the trial statistician who will be blinded to treatment allocation.

**Ethics and dissemination** The study has received full ethical approval. Study results will be disseminated via peer-reviewed publications, conference presentations and journal articles. Data obtained from the trial will enable clinicians and health service providers to make informed decisions regarding the efficacy of ACT for improving QoL for patients with MD.

**Trial registration number** NCT02810028.

### Strengths and limitations of this study

► Acceptance and Commitment Therapy for Muscle Disease is the first randomised controlled trial to test the efficacy of an Acceptance and Commitment Therapy-informed intervention for muscle diseases (MDs) against standard medical care (SMC).

► To our knowledge, this is the largest trial of a psychological intervention for MD that primarily targets quality of life.

► In order to improve access and increase uptake, the intervention is brief and designed to be delivered remotely (via email and telephone).

► Only a short follow-up period was included, and consequently, this study will not be able to investigate longer-term treatment effects. An intrinsic limitation to the study is the lack of comparable therapist's time and attention within the SMC alone group.

**Protocol version** V.11 (4 April 2017).

## INTRODUCTION
### Background and rationale

Muscle diseases (MDs) are a group of primary disorders of the muscle, the majority of which are chronic and progressive. They affect approximately over 70 000 people (children and adults) in the UK.[1] The diagnosis of a specific MD is based on several criteria including the pattern of the muscle weakness, the muscle biopsy appearances and in some cases genetic testing. Four MDs, namely limb girdle muscular dystrophy (LGMD), Becker's muscular dystrophy (BMD) (dystrophin deficiency), facioscapulohumeral muscular dystrophy (FSHD) and inclusion body

myositis (IBM), comprise a significant proportion of those seen in an adult muscle clinic. They are characterised by progressive limb muscle wasting and weakness that cause difficulties with mobility and physical functioning. Some MDs also involve bulbar muscles causing dysphagia, respiratory muscles causing respiratory compromise and cardiac muscle causing cardiac symptoms. As well as weakness, commonly reported symptoms include chronic pain, sleep disturbance and fatigue.[2 3] Unlike some MDs, these four MDs do not have additional multisystem or central nervous system involvement. At present, most MDs are without disease-modifying treatments or cures.

Typically, people with MD report reduced quality of life (QoL) and may experience increased levels of anxiety and depression.[4 5] A systematic review of 26 studies highlighted that, compared with controls, people with MD had compromised QoL in all areas of functioning.[5] The degree of physical disability is a determinant of QoL in MDs.[5] Therefore, clinical treatment is often directed at maintaining physical functioning. For example, physiotherapy and occupational therapy are applied to maximise remaining muscle function, and patients are monitored for any cardiac or respiratory problems that may need supportive treatment.

However, physical disability does not explain all the variance in QoL.[4 6 7] Psychological variables such as illness beliefs,[6 8] coping methods[9–11] and psychological flexibility[12] have been shown to account for variance in QoL. These findings are in line with a biopsychosocial model[13] where QoL is theorised to depend on the interaction of psychological factors with biological and social factors.

The relationship between QoL and psychosocial factors in MD provides a treatment opportunity. In the absence of a direct cure for MD, psychological interventions may offer a way to enhance QoL. Encouragingly, a recent small randomised controlled trial (RCT) of a traditional cognitive behavioural intervention for improving fatigue in FSHD showed a beneficial effect on fatigue and QoL.[14]

## Acceptance and Commitment Therapy

Dependent on treatment aims, several types of psychological interventions may be helpful for people with MD.[15] Acceptance and Commitment Therapy (ACT) is a newer form of cognitive-behavioural approach that focuses on improving a process called psychological flexibility.[16] Psychological flexibility is "[…] the capacity to persist or to change behaviour in a way that includes conscious and open contact with thoughts and feelings, appreciates what the situation affords, and serves one's goals and values".[17] In observational studies, psychological flexibility has been shown to be predictive of positive outcomes (QoL and mood) and better functioning across a range of challenging contexts, such as living with chronic illness and chronic pain. A 4-month prospective observational study with a group of people with MD observed that psychological flexibility was predictive of change in life satisfaction and anxiety.[18]

Given its focus on improving psychological flexibility, ACT may prove beneficial for those with MD. In ACT, a number of treatment techniques are used to improve psychological flexibility including therapist interaction, mindfulness and functional analysis. These aim to: (1) help individuals become more aware of thoughts and feelings and their relationship to, in this case, their MD; (2) broaden the range of behaviours (often to include open and willing responses) that can be applied when in the presence of difficult thoughts and feelings and (3) help them to flexibly choose those behaviours that help them to make progress on their overarching goals and values.

Several recent meta-analyses have suggested that ACT is an efficacious treatment for improving mood and QoL in mental health conditions[19 20] and chronic pain.[21] In chronic diseases, there is a growing evidence base,[12] with encouraging results for trials with cancer,[22] diabetes[23] and epilepsy[24] populations. However, to date, most studies of ACT for chronic diseases have been of a preliminary nature (ie, small, uncontrolled and/or lower quality).[12] Thus, an adequately powered, rigorous evaluation of ACT for improving QoL in chronic diseases will be a helpful addition to the literature.

## The selection of a guided self-help approach

The one comprehensive trial of a psychological intervention in MD[14] tested the efficacy of a 16-session traditional cognitive behavioural therapy for improving fatigue. While results were encouraging, on both fatigue and secondary outcomes such as QoL and mood, for several reasons (including the time spent travelling), uptake was low, and a proportion (24%) struggled to achieve even a minimal level of adherence to the intervention. This suggests that issues, perhaps mobility difficulties and treatment length, may hinder face-to-face treatment options for many with MD. A further problem with the development and implementation of any new treatment concerns the resource limitations common to many health service providers, including the National Health Service (NHS) in the UK.

A practical solution to this problem could be the implementation of an intervention delivered remotely through online and computerised self-guided resources aided by telephone or video calls. Such methods of delivery are increasingly feasible given that the majority of UK households now have web access. In addition, recent systematic reviews suggest that promising results can be obtained from web-based interventions for adults with chronic illnesses and depression.[25–27] Web-based interventions not only remove the burden of travel but also allow the completion of material at a convenient place, time and pace. Remotely delivered interventions can be a cost-effective approach for providing treatment, increasing its chances of wider implementation.

Given the potential benefits of using ACT within an MD population and taking into account potential barriers to participation, we developed, for this population, a brief

self-guided intervention that can be supported remotely via email and telephone contact with a therapist. We assessed this in a preliminary multiple-baseline case series with seven participants.[28] The results of this case series suggested that such an approach was practicable and acceptable, and that mood and QoL might respond to this treatment. Given this promising evaluation, we plan to conduct an RCT, to evaluate the efficacy of this intervention for improving QoL in people living with MD.

This protocol describes a multicentre, two-armed, parallel-group RCT to assess the efficacy of self-guided ACT in improving QoL in individuals with MDs. A total of 154 people referred from MD clinics, Muscular Dystrophy UK (MD-UK) and specific MD registries will be randomly allocated to receive either self-guided ACT plus standard medical care (SMC) or SMC alone. We hypothesise that the group who receive self-guided ACT plus SMC will report an improved QoL at the primary endpoint of 9 weeks follow-up compared with those receiving SMC alone.

## Objectives
### Primary objective
The primary objective is to evaluate the efficacy of ACT plus SMC on QoL compared with SMC alone at 9 weeks.

### Secondary objectives
1. To evaluate the impact of ACT plus SMC compared with SMC alone on secondary outcomes: mood, symptom impact domains, work and social adjustment and function at 9-week follow-up;
2. To assess the processes that may be responsible for mediating the treatment effect;
3. To evaluate participants' views of the intervention and factors that act as facilitators and barriers to participation via a nested qualitative study.

## Trial design
Acceptance and Commitment Therapy for Muscle Disease is an assessor-blind, multicentre, two-armed, parallel-group RCT.

## METHODS: PARTICIPANTS, INTERVENTIONS AND OUTCOMES
### Setting
A total of 154 participants will be recruited from UK NHS MD clinics, MD-UK and three disease-specific registries (FSHD, dysferlin and IBM). After obtaining consent, the baseline assessments will take place in MD clinics at King's College Hospital (NHS Foundation Hospital Trust), the Royal London Hospital (Barts Health NHS Trust) and University Hospital Southampton (NHS Foundation Trust). The remainder of the study including treatment will be conducted remotely via email, telephone or video calls.

## Eligibility criteria
### Inclusion criteria
1. Adults (aged 18 years and over) diagnosed with one of the following MDs that has been present for greater than 6 months:
2. one of the limb girdle muscular dystrophies LGMDs genetically or biopsy proven,
3. Becker's muscular dystrophy (BMD (dystrophin deficiency) with biopsy or genetic diagnosis,
4. Facioscapulohumeral muscular dystrophy (FSHD) diagnosed clinically with specific genetic abnormality in the subject or their family,
5. Inclusion body myositis (IBM) clinic-pathologically defined, clinically defined or probable IBM based on European Neuromuscular Centre research diagnostic criteria 2013,[29]
6. Potential participants must have access to the internet and a computer to receive the study materials;
7. Hospital Anxiety and Depression Scale (HADS) score of ≥8 for depression or ≥8 for anxiety.

### Exclusion criteria
1. Unstable complications of MDs including neuromuscular respiratory weakness or 'cardiomyopathy,
2. Major active comorbidities unrelated to MD (such as arthritis, respiratory disease, cardiovascular disease),
3. Current diagnosis of an active major mental health disorder likely to interfere with participation,
4. Current or recent participation in other treatment intervention studies (<4 weeks after completion),
5. Currently receiving psychological support or psychotherapy,
6. Inability to read English questionnaires,
7. Cognitive impairment that would prevent comprehension of ACT modules and questionnaires (as assessed by the Montreal Cognitive Assessment 5 min protocol[30]).

## INTERVENTIONS
### Acceptance and Commitment Therapy
The intervention is a remotely supported, guided self-help ACT programme (developed mainly by CDG and LM, informed by other sources).[16 22 31] It consists of four modules and corresponding audio files supported by five telephone support sessions with the trial therapist (table 1). The first three modules are expected to take the participant 1 hour 30 min each, and the final module 45 min to complete. This includes time to read the modules, listen to the audio files and work through tasks and exercises. During the first call, prior to receiving the first module (by either telephone or video call), the therapist will provide a 15-minute introduction then email the first module. The subsequent three phone calls will each last 30 min and will be followed by the next module. The telephone sessions provide participants with a chance to discuss the modules they have read, including their experiences of the exercises, and

**Table 1**  Summary of ACT modules

| | |
|---|---|
| Module 1: mindfulness and unhooking | ▶ Normalising difficult thoughts and feelings, given the context of MD, and associated struggles.<br>▶ Discussion of consequences from trying to get rid of unwanted thoughts and feelings and opportunities for alternative responses.<br>  1. Introducing other ways to interact with unwanted thoughts and feelings:Mindfulness or learning to be present in the moment and experience thoughts and feelings in a non-judgemental way.<br>  2. Unhooking from thoughts or learning ways to step-back from thoughts that often restrict actions or lead to avoidance.<br>▶ Homework 1: daily diary of mindfulness practice.<br>▶ Homework 2: identifying where one struggles and experimenting with other responses. |
| Module 2: follow your values | ▶ Identifying values and ways to pursue them with MD. Exploring the link between difficult thoughts and feelings and one's goals and values.<br>▶ Homework 1: continuation of daily mindfulness practice<br>▶ Homework 2: setting goals in the context of values, experimenting with taking the 'smallest possible step' which is consistent with one's values, noticing the thoughts and feelings that occur in the context of values activity. |
| Module 3: take an observer perspective | ▶ Experimenting with taking an observer perspective on one's experiences.<br>▶ Considering labels one attaches to oneself (especially given the context of MD). Noticing a choice over buying into labels and the impact of labels over one's behaviour.<br>▶ Introducing a more flexible approach to mindfulness<br>▶ Homework 1: continuation of daily mindfulness practice with addition of flexible attention and noticing the observer self<br>▶ Homework 2: continuation of valued activities |
| Module 4: recap, reflection and suggestions for staying committed | ▶ Review of homework tasks and skills learnt.<br>▶ Self-identification of effective and ineffective behaviour patterns.<br>▶ Goal planning and normalisation of set-backs with a compassionate approach to getting back on track. |

ACT, Acceptance and Commitment Therapy; MD, muscle disease.

receive help with difficulties encountered. A 15-minute phone call will follow the final module to conclude the treatment.

The clinical psychologist delivering the ACT support sessions will attend a 3-day training session prior to the start of recruitment. The first day will cover the rationale behind the study and insights into MD diagnosis, symptoms and impact on QoL (led by MR, a consultant neurologist). Details of the study protocol will be reviewed as well as the role of the therapist, standard operating procedures, recording and storage of the support sessions and confidential information, deviations, drop-outs and adverse event procedures (led by TC, professor of cognitive behavioural therapy). The ACT training including presentations, exercises and role plays will last 2 days. This will be led by LM, a clinical psychologist and expert in ACT, assisted by CDG, an ACT-trained clinical psychologist. Obstacles to patient engagement with ACT as well as the difficulties of delivering the sessions remotely and within a brief time frame will be discussed. The therapist will attend monthly supervision meetings with a clinical supervisor (LM) to develop their skills and to ensure therapy fidelity and adherence to trial protocol. The therapist's treatment competence will be periodically assessed with an ACT fidelity scale. This will inform monthly supervision sessions between therapist and clinical supervisor throughout the trial.

### Treatment fidelity

All ACT support sessions will be audio-recorded to assess treatment fidelity. At the end of the trial, a random sample of recordings will be analysed for overall fidelity by two independent researchers. A fidelity measure is currently being developed by CDG and LM. This comprises a basic therapeutic competence scale and an ACT-specific fidelity scale informed by guidelines provided by Plumb and Vilardaga.[32]

### Standard medical care

All participants will receive SMC, which involves the continuation of any current medical practices (without any psychological therapy). SMC would be (1) review of functional impairment arising from their muscle weakness and measures suggested to ameliorate the resulting disability with home adaptations and assistive devices, (2) monitoring for respiratory or cardiac complications and appropriate referral for their management if required, (3) recommendations for local physiotherapy, (4) recommendations for falls management where appropriate and (5) answering queries about their condition usually using information leaflets from the national organisation MD-UK or from disease support groups.

### Baseline measures

Participants will complete baseline questionnaires (see table 2) at the MD clinics with the research assistant

**Table 2** Screening and data collection across the trial: summary of key trial processes

| Process | Screening | Baseline | 3 weeks | 6 weeks | 9 weeks | Ongoing or during trial | Ref |
|---|---|---|---|---|---|---|---|
| Eligibility | X | | | | | | – |
| Medical confirmation | X | | | | | | – |
| HADS | X | X | X | X | X | | 37 |
| MoCA | X | | | | | | 28 |
| Participant details form | | X | | | | | – |
| INQoL | | X | X | X | X | | 33 |
| WSAS | | X | X | X | X | | 34 |
| HAQ-DI | | X | X | X | X | | 37 |
| IBM-FRS | | X | X | X | X | | 38 |
| AAQ-II | | X | X | X | X | | 40 |
| MASS | | X | X | X | X | | 41 |
| CAQ | | X | X | X | X | | 42 |
| Health events | | X | X | X | X | | – |
| ANA | | X | | | | | 31 |
| 6MTWT | | X | | | | | 31 |
| MMST | | X | | | | | 32 |
| PGIC | | | X | X | X | | 39 |
| Satisfaction rating | | | X | X | X | | – |
| Safety events (eg, AE, SAEs) | | | | | | X | – |
| Drop-out/withdrawal | | | | | | X | – |
| Therapist ratings | | | | | | X | – |

AAQ-II, Acceptance and Action Questionnaire; AE, adverse event; ANA, Adult Ambulatory Neuromuscular Assessment; CAQ, Committed Action Questionnaire; HADS, Hospital Anxiety and Depression Scale; HAQ-DI, Stanford Health Assessment Questionnaire Disability Index; IBM-FRS, inclusion body myositis functional rating scale; INQoL, Individualised Neuromuscular Quality of Life Questionnaire; MAAS, Mindfulness Attention Awareness Scale; MMST, Manual Muscle Strength Testing; MoCA, Montreal Cognitive Assessment 5-minute protocol; PGIC, Patient Global Impression of Change Scale; SAE, serious adverse event; WSAS, Work and Social Adjustment Scale; 6MTWT, 6-minute timed walk test.

(RA) and objective assessments with a physiotherapist. Participants will also complete a participant details form to provide demographic data as follows: age, gender, ethnicity, occupation status, education level, marital status, living arrangements and dependants. Clinical information will also be collected, including age of onset and duration of MD, additional diagnoses, experience of psychological therapies as well as prior treatment of anxiety or depression. Adverse events will be recorded at each assessment point of the trial.

The objective assessments conducted by physiotherapists at baseline are as follows:

► Adult Ambulatory Neuromuscular Assessment (ANA) is an adult version of the North Star Ambulatory Assessment[33] that measures motor function. Items are scored from 0 to 2, with an overall possible range of scores from 0 to 34. The higher the score, the better the patient can function. An additional item of the ANA that shall be used is the 6-minute timed walk test which measures mobility by counting the distance walked within a 6-minute time-frame along with pauses or stops before the time is complete. The higher the score, the greater the function of the patient.

► Manual Muscle Strength Testing (aka Manual Muscle Testing) measures the strength of 12 different muscles on both right and left from 0 to 5 (no contraction palpable–normal strength). There is a total possible range of scores from 0 to 120, with higher scores representing levels of strength closest to normal levels.[34]

## Outcome measures

The RA blind to participant allocation will be responsible for ensuring outcome measures are completed at 3, 6 and 9 weeks by all participants. The link to the measures will be sent to participants via email. If participants do not complete the questionnaires by the due date, they will be reminded by the RA. All time points will be taken into account during analysis, but the primary efficacy endpoint is the 9-week follow-up.

## Primary outcome

► The Individualised Neuromuscular Quality of Life Questionnaire (INQoL)—life area domains (section 2) is the primary outcome. The INQoL is a 10-domain, validated MD-specific questionnaire to measure QoL.[35] We will exclude the one domain that addresses the symptom of myotonia as none of the MDs being recruited has this symptom. This will leave nine domains for the purpose of this study. The primary outcome of this study constitutes section 2 of the INQoL which assesses five life area domains (activities, independence, social functioning, emotional functioning and body image). Participants respond using a seven-point Likert scale, with higher scores indicating greater symptom impact. The primary outcome measure for this study will therefore be the change in the overall QoL score, derived from these five life domain scores, from baseline up to 9 weeks.

## Secondary outcome measures

► INQoL symptom impact domains. This refers to categories within section 1 of the INQoL (weakness, pain and tiredness) and follows the same format as described above.

► Work and Social Adjustment Scale (WSAS) is a measure of impairment at work, home and social functioning.[36] The total score comprises the sum of responses to five questions, each of which is on a nine-point Likert scale. The possible range of scores is 0–40; with higher scores indicating more greatly impaired daily functioning.

► HADS is a self-report questionnaire to measure mood.[37] It consists of 14 items, relating to depression (seven items) and anxiety (seven items). Higher scores indicate either greater anxiety or depression. Each of the items is rated from 0 to 3; thus, the possible range of scores for each is 0–21, with a cut-off of 8 showing good sensitivity and specificity for possible cases.[38]

► Stanford Health Assessment Questionnaire Disability Index (HAQ-DI) is a measure of functional impairment and level of disability.[39] There are different ways to score the HAQ-DI. As with previous studies with MD populations,[18] the alternative scoring method shall be used which focuses on measuring disability not adaption by not incorporating use of aids and devices. The measure includes eight activity domains (dressing, arising, eating, walking, hygiene, reach, grip and activities). There are 20 questions in total, each scored from 0 to 3. All the domains are summed for a total score with a higher score representing a greater impairment in ability to function in daily life. As an additional measure of functional impairment, not treated as an efficacy outcome in the analysis, the IBM Functional Rating Scale[40] will be completed only by those participants with a diagnosis of IBM. This is a 10-point functional rating scale, with higher scores depicting greater functional impairment.

► Patient Global Impression of Change Scale (PGIC) is a single-item measure of a patient's impression of change over the study.[41] Participants rate their impression of change on a scale of 1–7 as follows: 'very much better', 'much better', 'a little better', 'about the same', 'a little worse', 'much worse' and 'very much worse'.

► Rating of satisfaction is a single-item measure of a patient's satisfaction with the outcome of the trial. Participants rate their satisfaction on a scale of 1–7 as follows: 'very satisfied', 'moderately satisfied', 'slightly satisfied', 'neither', 'slightly dissatisfied', 'moderately dissatisfied' and 'very dissatisfied'.

## Putative mediators and process variables

► Acceptance and Action Questionnaire (AAQ-II)[42] measures psychological flexibility, including experiential avoidance and psychological inflexibility. Each of the seven items are rated on a scale of 1–7, with higher scores representing greater experiential avoidance and immobility.

► Mindfulness Attention Awareness Scale (MAAS) measures dispositional open awareness of and attention to the present moment.[43] There are 15 items, each scored from 1 to 6. A mean of the 15 items is computed, with higher scores reflecting higher levels of dispositional mindfulness.

► Committed Action Questionnaire (CAQ) measures committed action, that is, goal-directed, flexible persistence.[44] Respondents rate the extent to which the eight items apply to them from 0 to 6 with a greater total score representing a greater general propensity to persist in goal-directed behaviour.

### Qualitative component

A nested qualitative study will explore patients' experience of the ACT intervention. This will help identify factors that facilitate or impede adherence and acceptability of the treatment and perhaps give insight into treatment mechanisms. Semi-structured interviews will be conducted post-intervention with approximately 15 participants (purposively sampled to encompass a mix of gender, ages and symptom severity). Interviewers will use topic guides with open-ended questions and prompts to elicit participants' accounts of their experiences. The interviews will be transcribed verbatim. Analysis will commence after the completion of the first interview in an iterative process to allow exploration of early insights in later interviews. Standard thematic analysis techniques will be applied,[45] for example, coding the data and identifying themes that capture key concepts and processes. The results will supplement quantitative results and inform any future implementation of the intervention.

### Participant timeline

Information is provided in the consort diagram (figure 1) and the screening and data schedule of assessments (table 2). The RA will contact potentially interested

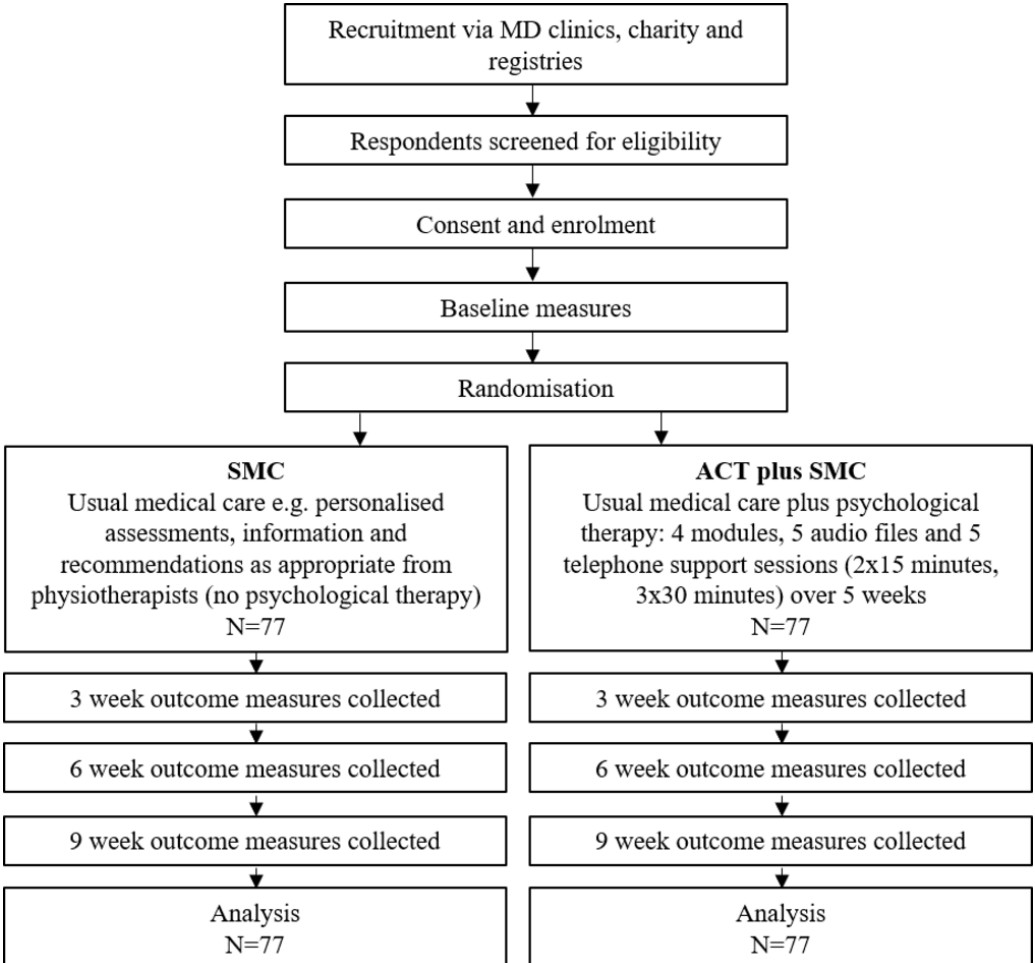

**Figure 1** Consort diagram for ACTMus. ACT, Acceptance and Commitment Therapy; ACTMus, Acceptance and Commitment Therapy for Muscle Disease; MD, muscle disease; SMC, standard medical care.

participants to ensure that they are fully informed and eligible to take part in the trial. If this is the case, they will be asked to sign a consent form, enrolled onto the study and provided with a unique identification number. The baseline assessments will be in person at one of the MD clinics with the RA and physiotherapist. Following baseline, the participant will be randomised and informed by the trial coordinator as to whether they will receive ACT plus SMC or SMC alone. The participant will complete follow-up measures at 3, 6 and 9 weeks online via an email link.

### Sample size
One hundred and fifty-four patients will be recruited in total, 77 per trial arm. This sample size will provide adequate power to detect a standardised mean difference of 0.5 (medium magnitude) in the primary outcome between the treatment and control groups at the end of intervention assessment (5% significance, 80% power), assuming attrition of 20%. Using data from Vincent *et al*,[35] this equates to between an 11.5-point and 15.8-point difference on each of the INQoL domains. While no information is currently available regarding clinically important differences for the INQoL, an effect size of this magnitude typically relates to a clinically important difference for health-related QoL instruments.[46]

### Recruitment
Participants will be recruited from MD clinics, MD-UK and MD registries. In MD clinics, consultants and physiotherapists will identify potentially eligible participants who will either be given an information pack consisting of an invitation letter, information sheet and reply slip to state their interest. MD-UK and MD registries will advertise the study and provide invitation packs. When someone states their interest, the RA will call them to provide further information, answer any questions and conduct screening. If they are eligible for the study (including medical confirmation from their health professional), then the participant will sign a consent form before being enrolled into the study.

### Patient and public involvement
MR's direct work with patients within an MD clinic led to the development of the trial. In developing the study material, a small pilot study of the self-guided intervention was tested and subsequently refined in response to participants' feedback by CG (Graham, 2012). Our patient representative has been actively involved at every

stage from assessing the intervention to reviewing the outcome measures. MD-UK, a patient support organisation, contributed to the development of the study and will help with the referral of patients into the study. MD-UK will also be involved in disseminating study results.

## METHODS: ASSIGNMENT OF INTERVENTIONS
### Allocation
Randomisation will be conducted by an independent randomisation service at the UK Clinical Research Collaboration registered at the King's Clinical Trials Unit. Randomisation will be at the level of the individual, using block randomisation with randomly varying block sizes stratified by recruiting centre. Allocation confirmation will be generated automatically and sent to unblinded research members, including the trial coordinator and clinical psychologist.

### Allocation concealment
A position of equipoise will be maintained, but blinding of the patients and therapist to the treatment condition will not be possible. The RA contacting participants and inputting data will be blind to group allocation to reduce the risk of introducing bias. To protect against unblinding, all follow-up questionnaires will be completed by participants on Bristol Online Survey (BOS) which is specifically designed for academic research and fully compliant with UK data protection law. All questionnaires will be identical for both arms and designed not to reveal treatment allocation. The trial statistician will be blind to treatment group when conducting the main data analysis.

## METHODS: DATA COLLECTION, MANAGEMENT AND ANALYSIS
### Data collection methods
Baseline data will be collected prior to randomisation and consist of self-report questionnaires and objective assessments with the physiotherapist. Follow-up data (3, 6 and 9 weeks) include self-report questionnaires that will be completed remotely. At each follow-up point, participants will be emailed a link to BOS along with their unique identification number. If participants are unwilling or unable to complete all the questionnaires, they will be given the option of going through them on the phone or a reduced questionnaires pack with the primary outcomes (INQoL).

### Data management
All trial data will be stored in line with the Data Protection Act 1998 and in compliance with Good Clinical Practice and King's College London data management procedures. Any identifying data will be anonymised through use of unique identification numbers separated from identifying data. Data will be entered onto password-protected trial databases either on a shared hard-drive accessible only to assigned team members or in a locked cabinet in a room only accessible to staff with designated access cards. All source documents will be retained for a period of 10 years following the end of the study.

## Statistical methods
Statistical analyses of the primary and secondary outcomes will be conducted after the database has been locked, with no interim analyses. All analyses will adopt the intention to treat principle. The main statistical analyses will estimate the difference in mean outcomes between patients randomised to ACT plus SMC and SMC alone at the primary endpoint of 9 weeks and at the two other postrandomisation time points. Group difference estimates, associated CIs and standardised effect sizes will be reported. Significance testing will be performed where the significance level will be 5% (two-sided). Sensitivity analyses will be used to assess the robustness of conclusions to missing outcome data using a pattern mixture model approach and to departures from randomised treatment in a per protocol analysis excluding those not receiving. Loss to follow-up, departures from randomised treatment and the prevalence of serious adverse events (SAEs) will be reported.

The primary outcome is the INQoL QoL score. Quality of life at all three postrandomisation time points (3, 6 and 9 weeks after randomisation) will be modelled simultaneously using a linear mixed-effects model. A random intercept will account for the non-independence of observations across individuals. Trial arm and indicator-coded time will be included as covariates along with arm by time interaction terms to allow the treatment effect to vary at each time point. In addition, included in the model will be the baseline level of the QoL score and a stratification indicator variable for centre. Missing data at follow-up assessments are allowed where at least one observation per participant is available under the missing at random assumption.

Secondary patient outcomes (3, 6 and 9 weeks after randomisation) relating to symptom severity and functioning (WSAS and HAQ-DI), distress (HADS), impression of change (PGIC) and putative mediators of the intervention (CAQ, AAQ-II and MAAS) will be analysed using linear mixed-effects models. Random effects for the intercept and time will be included in the model.

## METHODS: MONITORING
### Data monitoring
The study will be monitored and audited in accordance with King's College Hospital NHS Foundation Trust and King's College London procedures. All trial related documentation will be made available on request to the sponsors, Health Research Authority/Research Ethics Committee (REC) and other licensing bodies. Monitoring of this study to ensure compliance with Good Clinical Practice and scientific integrity will be managed by the study team and reviewed by an independent Trial Steering Committee (TSC) and Data Monitoring & Ethics Committee (DMEC).

### Harms/assessment of safety
Adverse events will be recorded at baseline, 3, 6 and 9 weeks. We define adverse events as "any clinical change,

disease or disorder experienced by the participant during their participation in the trial, whether or not considered related to the use of treatments being studied in the trial".[47] Adverse events will also include any other events that might have affected the health status of a participating patient (eg, increased work stress).

The team will also monitor SAEs, serious adverse reactions (SAR), suspected unexpected serious adverse reactions (SUSAR) and active withdrawals from treatment. An SAE constitutes any adverse event or reaction that: results in death or a new persistent or new significant disability or incapacity, is life threatening or requires hospitalisation, any new episode of deliberate self-harm or any other important medical condition which may jeopardise the participant and requires medical or surgical intervention to prevent one of the outcomes listed above.

An SAR can be defined as an SAE as a reaction to the intervention. An SUSAR is any SAR that is suspected to be caused by the intervention but was not expected. All SAEs, SARs and SUSARs will be reported immediately by the chief investigator to the study sponsor and REC. The TSC-DMEC will be responsible for the independent investigation of SAEs.

### Research ethics approval

Site-specific confirmation of capacity and capability has been given by local research and development departments. The study will comply with the principles of the Declaration of Helsinki (1996),[48] the International Conference for Harmonisation of Good Clinical Practice[49] guidelines and the Research Governance Framework for Health and Social Care Second Edition (2005). The trial is sponsored by King's College Hospital NHS Foundation Trust (lead) and King's College London and has been registered onto a publicly accessible database (registration no NCT02810028).

### Consent

Inclusion in the trial is appropriate only if potential participants are aged 18 years or older and are able to give written informed consent. All potentially eligible participants will be fully informed about the study procedures by a participant information sheet that states participation is voluntary, and that they are free to withdraw from the study at any time without giving a reason and without their care being affected.

### Declaration of interests

The authors declare that they have no competing interests.

### Dissemination

Summaries of findings will be offered to MD charities, registries and support groups as well as to the wider public. The results of the trial will be communicated to participants via a newsletter. Any results from the trial will enable informed decisions regarding the provision of care for patients with MD.

## CONCLUSION

Experiences of psychological distress and reduced QoL are common in people with MD. A brief, guided self-help ACT intervention with telephone support sessions has been designed for remote delivery, offering a potential solution to mobility issues inherent with MD and under-resourced healthcare services. This is the first study to evaluate whether ACT can help improve QoL for individuals with MD. It will also be one of the largest RCTs of ACT for improving outcomes in chronic diseases/long-term conditions. If the intervention proves to be efficacious, this treatment delivery could facilitate participation for those individuals whose impaired mobility prevents travelling to face-to-face sessions. As the main barrier to the adoption of any psychological intervention is the paucity of psychological services, the ACT intervention has been designed to be a self-guided programme requiring minimal support from highly qualified professionals.

**Author affiliations**
[1]Department of Neurology, King's College Hospital, London, UK
[2]Department of Psychology, Institute of Psychiatry, Psychology and Neuroscience, King's College London, London, UK
[3]Department of Psychological Medicine, Institute of Psychiatry, Psychology and Neuroscience, King's College London, London, UK
[4]Leeds Institute of Health Sciences, University of Leeds, Leeds, UK
[5]Royal London Hospital, Barts and the London MND Centre, London, UK

**Acknowledgements** We are grateful to Daniel Thomas (PPI representative) and MD-UK representatives who commented on several drafts of the study materials at different stages. We are also grateful to our physiotherapy collaborators (Joanna Reffin, Emily Jay, Sunitha Narayan, Jade Donnelly, Leslie Richards and Kelly Orr).

**Contributors** MRR, CDG, SN, LM, AR and TC obtained funding for the study. MRR, SN, CV, VE, LM, CDG, AR and TC contributed to the study protocol. MRR and TC are responsible for overall supervision. All authors read, contributed and approved the final manuscript.

**Funding** National Institute for Health Research (NIHR) Research for Patient Benefit grant (reference no PB-PG-061331085). Muscular Dystrophy UK have been partners from the commencement of the study and funded the trial therapist. The trial is sponsored by King's College Hospital NHS Foundation Trust and cosponsored by King's College London.

**Competing interests** None declared.

**Patient consent** Not required.

**Ethics approval** Health Research Authority (London-Camberwell St Giles Research Ethics Committee, 16/LO/0609).

**Provenance and peer review** Not commissioned; externally peer reviewed.

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
