## [Reviewer comments · BMJ Open]

ARTICLE DETAILS

TITLE (PROVISIONAL)	Acceptance and Commitment Therapy for MuScle disease (ACTMuS): protocol for a two-arm randomised controlled trial of a brief guided self-help ACT programme for improving quality of life in people with muscle diseases
AUTHORS	Rose, Michael; Norton, Sam; Vari, Chiara; Edwards, Victoria; McCracken, Lance; Graham, Christopher; Radunovic, Aleksandar; Chalder, Trudie

VERSION 1 – REVIEW

REVIEWER	Chiara A M Spatola Catholic University of the Sacred Heart, Milann Istituto Auxologico Italiano IRCCS, Milan, Italy
REVIEW RETURNED	14-Mar-2018

GENERAL COMMENTS	Summary This is a study protocol for a two-arm, multicenter, parallel-group RCT of a brief guided self-help ACT program for improving quality of life in people with muscle diseases. The main aim of the study is to evaluate the efficacy of the ACT treatment plus standard medical care for the improvement of QOL compared with SMC alone at 9-weeks follow-up. Secondary outcomes include: mood, symptom impact domains, work and social adjustment and function at 9-week follow-up. Moreover, the study aims to assess possible mediation effects and to evaluate participants' views of the intervention, as well as possible factors acting as facilitators and barriers to participation (via a nested qualitative study). Strengths a. This is a very elegant study protocol that is novel from several perspectives: it is the first study to evaluate the efficacy of an ACT-based intervention on QoL in individuals with muscle disease. It will also be one of the largest randomised controlled trials testing the efficacy of an ACT-based intervention to improve outcomes in chronic conditions. Moreover, the inclusion of telephone support sessions, designed for remote delivery, offers a promising solution to mobility issues inherent with MD. b. The introduction presented a thoughtful summary of literature to date and solid rationale for the study, including its novelty. c. The assessment procedure is well planned. The baseline assessment measures include records of adverse events and objective assessments conducted by physiotherapists. The process outcome measures include process variables and possible mediators of the treatment effects. d. The methodology and the statistical analyses seem appropriate to answer the study questions and very clearly presented. The sample
---

	size (154 patients) has been adequately planned by power analyses. Analyses are planned to be intent to treat. e. The inclusion of a qualitative component for exploring patients' experience of the ACT program is a strength of the study Weaknesses Overall, this study was very well reasoned and presented. I have only minor comments below that I think might strengthen the manuscript: a. In page 9 line 37, the authors included, among the exclusion criteria "currently receiving psychological support or psychotherapy". However, since an inclusion criterion is the presence of anxious or depressive symptomatology, it would also be appropriate to control whether patients receive or not pharmacological treatments for anxiety and depression. The participants assigned to the control condition, who don't receive any specific treatment for anxiety or depression, may be more prone to seek other treatments (not psychological therapies) to enhance their mental health (e.g. asking their doctors for drugs prescriptions, contacting patient support organizations, ...). I would suggest to keep track of any alternative intervention or practice that participants in both arms might receive to enhance their mental health. In case the authors are unable to exclude the presence of threats to the internal validity of the study, this should be acknowledged in the limitations. b. More details about the standard medical care should be provided. It seems that SMD is a kind of minimally-enhanced non-study care (as per definition of this study: 10.1097/PSY.0b013e318218e1fb), in which participants are given information when required and then simply allowed to continue their previous care, that is provided by doctors who are independent from the study. However, this is not entirely clear and the authors should state who provides SMD. c. It is unclear how personalized information given at baseline is standardized. The authors should clarify whether it is based on a set of pre-defined information which is shared by all physiotherapists of the clinics, on standardized material (e.g. a booklet) or other. In addition, since recruitment is based on registries, participants are likely to be resident in different regions of the UK and it is unclear how information about patient support organizations were standardized. d. Since SMD is a minimally enhanced non-study care, some threats to the internal validity of the study cannot be excluded, some of them are listed in this study: 10.1097/PSY.0b013e318218e1fb. Moreover, the presence of a control group which does not receive a comparable treatment does not allow to distinguish between specific (i.e. linked to the ACT model) and nonspecific treatment effects of the programme (due to monitoring and relation with the therapist). Providing a different psychological treatment (i.e. standard CBT or psychoeducation) to the control group can be a possible solution. In case the authors cannot address this issue, it should be mentioned among the limitations of the study. e. The HADS is included in the study as a secondary outcome measure and as a screening tool to identify patients with depressive (score ≥ 8) or anxious (score ≥ 8) symptoms. Several studies (10.1016/j.jpsychores.2011.06.008, 10.1016/j.jpsychores.2012.10.010) have shown that the HADS has a volatile factorial structure and that its subscales should not be interpreted separately. Rasch analyses found contrasting results and a study on patients with motor neurone disease found that the original HADS had a poor fit to a Rasch model and suggested to use a modified version of its subscales (10.1186/1477-7525-9-82).
--	--

	Other questionnaires have a more stable factorial structure and can be used for screening purposes, e.g. the PHQ-9 (10.1046/j.1525-1497.2001.016009606.x, see its use in multiple sclerosis 10.1037/a0035919) and the GAD-7 (10.1001/archinte.166.10.1092). f. Please correct a typo in page 12, line 18 : “(7 items). . Higher scores” and the header of page 9, line 3 (“Method” is used instead of “Methods”)
REVIEWER	Peter Simpson University College Of Osteopathy, Borough High Street, London, UK; 275 Borough High Street, London SE1 1JE, UK
REVIEW RETURNED	24-Mar-2018
GENERAL COMMENTS	Thank you for the opportunity of reading this manuscript. There are some grammatical errors highlighted in the attached pdf. These are mainly in the introduction and on page 11 lines 23-24. I was unable to view figure 1 i.e. the consort diagram. - The reviewer provided a marked copy with additional comments. Please contact the publisher for full details.

VERSION 1 – AUTHOR RESPONSE

Thank you for your helpful revisions for this manuscript. Please see below for our responses and description of how we have revised the manuscript accordingly.

a. In page 9 line 37, the authors included, among the exclusion criteria “currently receiving psychological support or psychotherapy”. However, since an inclusion criterion is the presence of anxious or depressive symptomatology, it would also be appropriate to control whether patients receive or not pharmacological treatments for anxiety and depression. The participants assigned to the control condition, who don’t receive any specific treatment for anxiety or depression, may be more prone to seek other treatments (not psychological therapies) to enhance their mental health (e.g. asking their doctors for drugs prescriptions, contacting patient support organizations, ...). I would suggest to keep track of any alternative intervention or practice that participants in both arms might receive to enhance their mental health. In case the authors are unable to exclude the presence of threats to the internal validity of the study, this should be acknowledged in the limitations.

Thank you. We do monitor this as within our health events questionnaire (table 2: page 14, line 14), it asks “Please describe any other changes in your health status (including medication) over the past 3 weeks”. Both treatment and control groups fill in this questionnaire throughout the study.

b. More details about the standard medical care should be provided. It seems that SMD is a kind of minimally-enhanced non-study care (as per definition of this study: 10.1097/PSY.0b013e318218e1fb), in which participants are given information when required and then simply allowed to continue their previous care, that is provided by doctors who are independent from the study. However, this is not entirely clear and the authors should state who provides SMD.

Thank you for raising this, it is indeed the case that SMD are given information if required as part of their routine care. Within the trial, as stated p11 line 5, physiotherapists provide the SMC within the trial. Further clarification of standard medical care has been added to the methods section – under Standard Medical Care (SMC) subheading (page 11, line 3-7).

c. It is unclear how personalized information given at baseline is standardized. The authors should clarify whether it is based on a set of pre-defined information which is shared by all physiotherapists of the clinics, on standardized material (e.g. a booklet) or other. In addition, since recruitment is based

on registries, participants are likely to be resident in different regions of the UK and it is unclear how information about patient support organizations were standardized.

We agree that this requires some clarification as we cannot say with certainty that the personalised information is standardised in all clinics. However, the expected care would be 1) review of functional impairment arising from their muscle weakness and measures suggested to ameliorate the resulting disability with home adaptations and assistive devices, 2) monitoring for respiratory or cardiac complications and appropriate referral for their management if required, 3) recommendations for local physiotherapy, 4) recommendations for falls management where appropriate, 5) answering queries about their condition usually using information leaflets from the national organisation Muscular Dystrophy UK. We therefore we do believe there will be a certain level of consistency.

To make this clearer within the paper, we have modified the text to make the nature of Standard medical care clear (page 11, line 3-7).

d. Since SMD is a minimally enhanced non-study care, some threats to the internal validity of the study cannot be excluded, some of them are listed in this study: 10.1097/PSY.0b013e318218e1fb. Moreover, the presence of a control group which does not receive a comparable treatment does not allow to distinguish between specific (i.e. linked to the ACT model) and nonspecific treatment effects of the programme (due to monitoring and relation with the therapist). Providing a different psychological treatment (i.e. standard CBT or psychoeducation) to the control group can be a possible solution. In case the authors cannot address this issue, it should be mentioned among the limitations of the study.

Thank you for highlighting this point. As the study design cannot be changed at this point, as per your suggestion this has now been added as a limitation to the study within the paper (within the strengths and limitations section).

e. The HADS is included in the study as a secondary outcome measure and as a screening tool to identify patients with depressive (score ≥ 8) or anxious (score ≥ 8) symptoms. Several studies (10.1016/j.jpsychores.2011.06.008, 10.1016/j.jpsychores.2012.10.010) have shown that the HADS has a volatile factorial structure and that its subscales should not be interpreted separately. Rasch analyses found contrasting results and a study on patients with motor neurone disease found that the original HADS had a poor fit to a Rasch model and suggested to use a modified version of its subscales (10.1186/1477-7525-9-82). Other questionnaires have a more stable factorial structure and can be used for screening purposes, e.g. the PHQ-9 (10.1046/j.1525-1497.2001.016009606.x, see its use in multiple sclerosis 10.1037/a0035919) and the GAD-7 (10.1001/archinte.166.10.1092).

The HADS does indeed have a slightly complicated factor structure, which is partly due to the presence of a strong underlying general distress factor. We are aware of this issue since the HADS is an instrument commonly used by members of this group. Indeed the first author of one of the papers cited by the reviewer is the trial statistician for ACTMuS (10.1016/j.jpsychores.2012.10.010). The paper cited that provides modified versions of the HADS-A and HADS-D (10.1186/1477-7525-9-82) is based on a relatively small sample (for psychometric analyses) of people with motor neurone disease, which may not generalise to people with muscular dystrophy. Therefore, we plan to use the original scales as intended by Zigmond & Snaith including the total score. Since this trial involves a randomised comparison of groups any confounding in the instrument due to DIF will not bias between group differences but will add to measurement error. As such the issue is one of power rather than confounding. We will however carefully consider the psychometric properties of the HADS and other self-report scales using baseline data and make a-priori adjustments where necessary. We will pay particular attention to the items "I feel as if I am slowed down" and "I feel restless as if I have to be on the move", since these have consistently shown to be problematic in people with physical impairments.

f. Please correct a typo in page 12, line 18 : "(7 items). . Higher scores" and the header of page 9, line 3 ("Method" is used instead of "Methods") OK

Thank you to both reviewers for noticing typos and grammatical errors, these have now been amended.

VERSION 2 – REVIEW

REVIEWER	Chiara A. M. Spatola Catholic University of the Sacred Heart, Milan, Istituto Auxologico Italiano IRCCS, Milan, Italy
REVIEW RETURNED	17-Jun-2018
GENERAL COMMENTS	The authors adequately and clearly addressed all the issues.